# Use of Insect Exclusion Row Cover and Reflective Silver Plastic Mulching to Manage Whitefly in Zucchini Production

**DOI:** 10.3390/insects14110863

**Published:** 2023-11-09

**Authors:** Thiago Rutz, Timothy Coolong, Rajagopalbabu Srinivasan, Alton Sparks, Bhabesh Dutta, Clarence Codod, Alvin M. Simmons, Andre Luiz Biscaia Ribeiro da Silva

**Affiliations:** 1Department of Horticulture, Auburn University, 101 Funchess Hall, Auburn, AL 36849, USA; trd0017@auburn.edu; 2Department of Horticulture, University of Georgia, 1111 Miller Plant Sciences, 120 Carlton Street, Athens, GA 30602, USA; tcoolong@uga.edu; 3Department of Entomology, University of Georgia, 1109 Experiment Street, Griffin, GA 30223, USA; babusri@uga.edu (R.S.); asparks@uga.edu (A.S.); 4Department of Plant Pathology, University of Georgia, 3250 Rainwater Road, Tifton, GA 31793, USA; bhabesh@uga.edu (B.D.); cbc71306@uga.edu (C.C.); 5USDA-ARS, U.S., Vegetable Laboratory, Charleston, SC 29414, USA; alvin.simmons@usda.gov

**Keywords:** *Bemisia tabaci*, biomass accumulation, *Cucurbita pepo*, pest management, total yield

## Abstract

**Simple Summary:**

Sweet potato whitefly is a major pest in zucchini production due to the direct and indirect damage it causes to this crop. The present study evaluated reflective silver plastic mulch and an insect row cover as alternative methods to manage whiteflies in zucchini production in the southeastern U.S. Field experiments indicated reduced whitefly numbers under reflective silver plastic mulching and insect row cover treatments. These methods also improved plant growth and yield compared to conventional white plastic mulch and no-cover treatments. In conclusion, these alternative methods can be considered ready-to-use integrated pest management practices for growers.

**Abstract:**

The challenges that sweet potato whitefly (*Bemisia tabaci*) creates for vegetable production have increased in the southeastern U.S. Growers must use intensive insecticide spray programs to suppress extremely high populations during the fall growing season. Thus, the objective of this study was to evaluate the use of a reflective plastic mulch and an insect row cover as alternative methods to the current grower practices to manage whiteflies in zucchini (*Cucurbita pepo*) production. Field experiments were conducted with a two-level factorial experimental design of cover and plastic mulch treatments arranged in a randomized complete block design, with four replications in Georgia in 2020 and 2021, and in Alabama in 2021. Cover treatments consisted of an insect row cover installed on zucchini beds at transplanting and removed at flowering and a no-cover treatment, while plastic mulch treatments consisted of reflective silver plastic mulching and white plastic mulching. During all growing seasons, weather conditions were monitored, whitefly populations were sampled weekly, zucchini biomass accumulation was measured at five stages of crop development, and fruit yield was determined at harvesting. Warm and dry weather conditions early in the growing season resulted in increased whitefly populations, regardless of location and year. In general, the reflective silver plastic mulching reduced whitefly populations compared to the conventional white plastic by 87% in Georgia in 2020, 33% in Georgia in 2021, and 30% in Alabama in 2021. The insect row cover treatment reduced whitefly populations to zero until its removal. Consequently, zucchini plants grown with the insect row cover and reflective silver plastic mulching had an increased rate of biomass accumulation due to the lower insect pressure in all locations. Zucchini grown using silver reflective plastic mulch and row covers had an overall increase of 17% and 14% in total yield compared to white plastic mulch and no-cover treatments, respectively. Significant differences in yield among locations were likely due to severe whitefly pressure early in the fall season, and total yields in Georgia in 2020 (11,451 kg ha^−1^) were 25% lower than in Georgia in 2021 (15,177 kg ha^−1^) and in Alabama in 2021 (15,248 kg ha^−1^). In conclusion, silver plastic mulching and row covers reduced the whitefly population and increased biomass accumulation and total yield. These treatments can be considered ready-to-use integrated pest management practices for growers.

## 1. Introduction

Sweet potato whitefly (*Bemisia tabaci*) is one of the most significant challenges facing vegetable production during the fall months in the southeastern U.S. The direct damage by the insect to vegetable crops and the whitefly’s role as a vector of many plant viruses can lead to significant yield losses [1]. The high insect pressure on vegetable crops is primarily due to the region’s special characteristics and row crop production overlap. Crops such as yellow squash/zucchini (*Cucurbita pepo*), cucumber (*Cucumis sativus*), snap beans (*Phaseolus vulgaris*), tomato (*Solanum lycopersicum*), cotton (*Gossypium hirsutum*), and others are all grown in proximity throughout the southeastern U.S. in the fall months. This may result in the widespread distribution of whiteflies among many regional crops [2].

Consequently, growers rely heavily on insecticide spray programs to control whitefly populations, which negatively impact production costs and may increase environmental concerns. Furthermore, the elevated insecticide exposure has increased the resistance of whiteflies to key chemical control agents, such as organophosphates, carbamates, pyrethroids, neonicotinoids, diamides, and insect growth regulators [3,4]. Sustainable crop management practices are required to help growers with effective integrated pest management (IPM).

Alternative crop management practices to control virus vectors, such as aphids (*Aphis* sp.), thrips, and even whiteflies, have been continuously evaluated on specialty crops [5,6,7,8,9]. A few examples of IPM strategies against whiteflies include the use of silver reflective plastic mulch, which was previously reported to reduce the insect numbers on snap beans [10], watermelons [11], tomatoes [12,13,14,15], and zucchini. Combining a reflective plastic mulching with imidacloprid resulted in a three-fold reduction in whitefly numbers on zucchini compared to those with a white plastic mulch control. In addition, it also reduced the incidence of cucurbit leaf crumple virus (CuLCrV)-infected zucchini plants [5].

The use of insect row covers has also been successfully used to protect plants against whiteflies in cantaloupe (*Cucumis melon*) [16,17], tomato [18,19], and zucchini [9,20,21]. Row covers combined with insect growth regulators reduced fruit damage and increased fruit size, weight, and quality in zucchini plants due to the reductions in adults, eggs, and pupae/nymphs per leaf compared with those in no-cover treatments [9]. The temporary pest exclusion systems that separate insects from host plants provide short-term solutions to insect damage and avoid an infestation in the critical stages of crop development [22].

The effectiveness of silver reflective plastic mulching and row covers in managing insect pests for vegetable production has been previously reported; however, their impact on crop development (i.e., zucchini) during whitefly management in the southeastern U.S. remains poorly studied. Thus, the objective of this study was to evaluate the use of reflective plastic mulching and insect row cover as alternative methods to the current grower practices against whiteflies in zucchini production during the fall season.

## 2. Materials and Methods

### 2.1. The Experimental Design and Crop Management

Field experiments were conducted on a commercial vegetable farm in Ty Ty, Georgia, USA, in 2020 (31°44′59″ N, 83°59′04″ W) and 2021 (31°41′59″ N, 83°78′59″ W), and at the Wiregrass Research and Extension Center from Auburn University, located in Headland, AL, USA (31°21′11″ N, 85°19′17″ W), in 2021.

In each location, a two-level factorial experimental design with the type of plastic mulch and row cover treatments was arranged in a randomized complete block design (r = 4). Plastic mulching treatments consisted of using white plastic mulching (Vaporsafe RM, Raven Industries, Sioux Falls, SD, USA), which is the standard practice for fall vegetable production in the region, or silver reflective plastic mulching (metalized low-density polyethylene 0.2 mm, 2.0 OD, Intergro, Clearwater, FL, USA). Cover treatments included an insect row cover treatment, installed in a low gothic tunnel shape (1.2 m tall, 1.8 m wide) with white polypropylene fabric (Agribon AG-15, Berry Global, Evansville, IN, USA) in each bed and a no-cover treatment. 

Zucchini seeds, cultivar Paycheck (Syngenta US, Greensboro, NC, USA), were planted into 200-cell trays filled with soilless media on 25 August 2020 and 26 August 2021 in Georgia, and 2 August 2021 in Alabama. Seedlings were greenhouse-grown until transplanting on 9 September 2020 and 13 September 2021 in Georgia, and 19 August 2021 in Alabama. Plants were grown on 15 cm tall, raised beds spaced 1.8 m center-to-center, with an in-row plant spacing of 30 cm. Cover treatments were applied at transplanting and removed at the first sign of anthesis, which was 20 days after transplanting (DAT) in Georgia 2020, 21 DAT in Georgia 2021, and 18 DAT in Alabama 2021. During the entire growing season, zucchini plants receiving the no-cover treatment were sprayed weekly with 205 g/ha of flupyradifurone (Sivanto 200 SL; Bayer CropScience, Research Triangle Park, NC, USA), 30 g/ha of pyriproxyfen (Knack; Valent, Walnut Creek, CA, USA), or 150 g/ha cyantraniliprole (Exriel; DuPont, Wilmington, DE, USA) to control whiteflies. Plots with the insect row cover treatments were not sprayed during the cover period but received the same spraying program as the no-cover treatment after cover removal. Crop management practices of fertilizer application, irrigation events, and disease and weed control were similar in all treatments and followed the University of Georgia Extension Cooperative recommendations [23]. 

### 2.2. Weather Parameters and Data Collection 

Weather conditions of maximum, minimum, and average air temperature, and rainfall events were recorded daily in all locations using the closest weather station within the Georgia Automated Weather Network [24] in Georgia and the Auburn University Mesonet in Alabama [25].

During the growing season, yellow sticky pest monitoring cards (7.6 × 12.7 cm; BASF, Research Triangle Park, NC, USA) were used to monitor the whitefly population weekly in each experimental unit at a density of 1794 traps per hectare. Yellow pest monitor cards were installed 15 cm above the ground and vertically oriented. The numbers of whiteflies were counted in an area of 77.4 cm^2^ in the center of each card. The number of whiteflies per trap was transformed to 1 m^2^. 

Above-ground plant tissue samples were collected at six points during crop development. Samples comprised two representative plants of each plot and were oven-dried at 65.5 °C until constant weight was reached. Subsequently, a dry biomass accumulation logistic curve was fitted by adapting the following equation [26]:(1)$$\mathrm{Crop}\;\mathrm{biomass}\;\mathrm{accumulation}=\frac{NM}{1+e^{-k(x-l)}}$$
where *NM* is the maximum crop dry biomass accumulation, *k* is the crop dry biomass accumulation rate constant, and *l* is the days to half biomass accumulation. 

Zucchini fruits were harvested when they met commercial standards for crop maturity and initiated at 27 DAT in all locations. Fruit was harvested thrice weekly for 4 weeks and 12 harvests. By the end of the last harvest, the total yield was estimated.

### 2.3. Statistical Analyses

All data were analyzed using linear mixed techniques implemented in SAS PROC GLIMMIX (SAS/STAT 9.4; SAS Institute Inc., Cary, NC, USA). Whitefly counts and dry biomass accumulation over time were analyzed as repeated measures, accounting for the correlation structure induced through the repeated sampling. According to the smallest Akaike’s information criterion, the variance-covariance matrix was adequately modeled as an ante-dependence structure of order 1. The total yield of zucchini was analyzed using treatments individually and their interactions as fixed effects. When the F value of the analysis of variance was significant, multiple mean comparisons were performed using Tukey’s test with a *p*-value of 0.05. 

## 3. Results

### 3.1. Weather Conditions

Daily average air temperatures were similar across the three locations and decreased with plant development (Figure 1). Average air temperatures during the study period were 23.6 °C in Georgia 2020, 22.9 °C in Georgia 2021, and 27.5 °C in Alabama 2021. Rainfall accumulation was similar among the three locations, totaling 183.3 mm in Georgia 2020, 191.4 mm in Georgia 2021, and 194.6 mm in Alabama 2021. Mainly, rainfall events were concentrated in the early season in Georgia 2020 and Georgia 2021 but more evenly distributed across the growing season in Alabama 2021.

### 3.2. Whitefly Population

For all growing seasons, whitefly populations were significantly impacted by the main effect of plastic mulching during crop development and the main effect of row cover during crop development (Figure 2). Whitefly populations were the highest during the early season in Georgia 2020 and Georgia 2021 but the highest during the late season in Alabama in 2021. Regardless of location, the silver plastic mulching reduced the number of whiteflies more than the white plastic mulching throughout the growing season, while the row cover treatments decreased the number of whiteflies compared to those in the no-cover treatment early in the season. On average, the average whitefly population for the silver reflective plastic mulching was 3 whiteflies per trap in Georgia 2020 (Figure 2A), 6 whiteflies per trap in Georgia 2021 (Figure 2B), and 12 whiteflies per trap in Alabama 2021 (Figure 2C). The whitefly population for the white plastic mulching averaged 24 whiteflies per trap in Georgia 2020, 9 whiteflies per trap in Georgia 2021, and 17 whiteflies per trap in Alabama 2020 (Figure 2D–F).

At a given sampling time, the number of whiteflies in the insect row cover treatment was significantly lower than that in the no-cover treatments until cover removal. Overall, the whitefly population in the insect row cover treatment until cover removal averaged 2 whiteflies per trap in Georgia 2020 (Figure 2D), 1 whitefly per trap in Georgia 2021 (Figure 2E), and 12 whiteflies per trap in Alabama 2021 (Figure 2F). Conversely, the whitefly population in the no-cover treatment averaged 26 whiteflies per trap in Georgia 2020, 13 whiteflies per trap in Georgia 2021, and 17 whiteflies per trap in Alabama 2020 (Figure 2D–F). After cover removal, the whitefly populations were not significantly different between the cover treatments.

### 3.3. Zucchini Plant Development and Total Yield

Dry biomass accumulation data were used to determine the maximum crop dry biomass accumulation, crop dry biomass accumulation rate, and days to half biomass accumulation for plastic mulching and row cover treatments within each location (Figure 3).

The estimated maximum crop dry biomass accumulation was higher for the reflective silver plastic mulching (NM = 3318.8 kg ha^−1^) than for the white plastic mulching (NM = 2713.2 kg ha^−1^) when zucchini was grown in Georgia in 2021 (Figure 3B). In contrast, white plastic mulch had a higher crop dry biomass accumulation when mulch treatments were compared in Georgia 2020 (NM = 2042.6 kg ha^−1^ for the reflective silver plastic mulching and NM = 2288.9 kg ha^−1^ for the white plastic mulching) (Figure 3A) and Alabama 2021 (NM = 2733.3 kg ha^−1^ for the reflective silver plastic mulching and 3075.8 kg ha^−1^ for white plastic mulching) (Figure 3C). Nevertheless, the crop dry biomass accumulation rate was higher with reflective silver plastic mulching, and this mulching reduced the time zucchini plants needed to reach half of the dry biomass in Georgia 2020 (k = 0.3028 and l = 27) and Georgia 2021 (k = 0.2495 and l = 32 days) compared to that with white plastic mulching in Georgia 2020 (k = 0.2044 and l = 29 days) and Georgia 2021 (k = 0.1891 and l = 31 days). In Alabama 2021, the crop dry biomass accumulation rate was lower for the reflective silver plastic mulch (k = 0.2659) than for the white plastic mulch treatment (k = 0.3003), and the time for plants to reach half of the dry biomass accumulation was similar between the plastic mulching treatments.

The response of zucchini plants grown under row cover treatments (Figure 3D–F) indicated that the maximum crop dry biomass accumulation was higher in the covered treatment in Georgia 2020 (NM = 2343.6 kg ha^−1^) and Georgia 2021 (NM = 3120.3 kg ha^−1^) than in the no-cover treatment in Georgia 2020 (NM = 1916.7 kg ha^−1^) and Georgia 2021 (NM = 2989.5 kg ha^−1^). In contrast, the row cover treatment had a reduced estimated maximum crop dry biomass accumulation (NM = 2708.2 kg ha^−1^) compared to that of the no-cover treatment (NM = 3091.5 kg ha^−1^) in Alabama 2021. In addition, the covered treatment had greater crop dry biomass accumulation rate estimates in Georgia 2020 (k = 0.2044) and Georgia 2021 (k = 0.2263) than the no-cover treatment in Georgia 2020 (k = 0.1740) and Georgia 2021 (k = 0.1887). In Alabama 2021, the crop dry biomass accumulation rate was lower for the covered treatment than in the no-cover treatment (k = 0.2525 and k = 0.3261, respectively). The days to half of dry biomass accumulation estimation in Georgia 2020 (l = 29 days) and Alabama 2021 (l = 33 days) was shorter under the covered treatment than in the no-cover treatment (l = 31 days and l = 35 days, respectively). In Georgia in 2021, the days to half of dry biomass accumulation estimation was similar for both cover treatments (l = 32).

Fruit total yield was significantly impacted by location, plastic mulching, and cover treatments (Table 1). For the main effect of location, the total yield was higher in Alabama 2021 (15,248 kg ha^−1^) and Georgia 2021 (15,177 kg ha^−1^) than in Georgia 2020 (11,451 kg ha^−1^). The total yield for zucchini grown under the silver reflective plastic mulching (15,246 kg ha^−1^) was higher than for zucchini plants grown under white plastic mulching (12,672 kg ha^−1^). Ultimately, the total yield was higher for plants grown under the row cover treatment (15,030 kg ha^−1^) than those grown without row covers (12,887 kg ha^−1^).

## 4. Discussion

Daily air temperatures during the early part of the production season were within the optimum range of 20 to 30 °C for whitefly development at the three locations studied, which may have resulted in ideal conditions for the growth and reproduction of whiteflies [27,28,29,30]. Rainfall events likely harmed whitefly populations [30]; the average whitefly population fell from 50 whiteflies per trap to 10 whiteflies per trap following a single rainfall event of 96 mm at 18 DAT in Georgia in 2020. Similarly, rainfall events in Georgia in 2021 (from 0 to 10 DAT) and in Alabama in 2021 (from 0 to 16 DAT) totaled 45 mm and 97 mm, respectively, which may have suppressed whitefly populations early in the season at those locations and in those years. The negative impact of rainfall events on whitefly populations has previously been reported in the southeastern U.S. [31]. 

In our study, the reflective silver plastic mulching and insect row cover treatments reduced whitefly populations, maximized dry biomass accumulation, and increased total fruit yields. The reflective silver plastic mulching treatment reduced the number of whitefly adults captured on sticky traps by 87% in Georgia 2020, 33% in Georgia 2021, and 30% in Alabama 2021 compared to white plastic mulching. Previous studies have reported similar results, with zucchini plants grown in living buckwheat (*Fagopyrum esculentum*) mulch and silver reflective mulching having lower whitefly populations than when grown in white plastic mulch. The authors also suggested that using living or reflective mulch alone or combined with imidacloprid can reduce whitefly populations by 70% on squash plants compared to those on non-treated plants grown on white plastic mulch [5]. Studies conducted with other crops, such as watermelon (*Citrullus lanatus*) and tomato (*Solanum lycopersicum*) [11,12], have reported lower whitefly populations when using reflective silver plastic compared to when using black plastic mulch. Silver plastic mulching may affect the whiteflies’ phototactic responses to specific wavelengths of reflected light [32]. Under visible wavelengths, the phototactic behavior of the greenhouse whitefly (*Trialeurodes vaporariorum*) was studied, and the violet and orange spectra attracted the lowest and the highest number of whiteflies, respectively [33]. Reflective silver plastic mulching is known to reflect light in the blue (400 to 500 nm) and the near ultraviolet (395 nm) regions [13], which explains the lower number of whiteflies being attracted to plants grown in silver reflective plastic mulching. 

Insect row covers are a temporary pest exclusion system that reduced the whitefly populations on zucchini plants to zero until their removal. A previous study reported similar results, with row covers reducing the mean densities of whiteflies recorded on zucchini plants, resulting in a higher total yield per plant than uncovered pesticide-treated plants [21]. Because row covers exclude all pests, total yields increased due to the exclusion of not only whitefly but also pickleworm, *Diaphania nitidalis* (Stoll), melonworm, *Diaphania hyalinata* (L.), and several colonizing aphid species from zucchini plants commonly reported in the southeastern U.S. [20]. It is still important to highlight that insect-excluding row covers must be removed at anthesis to allow flower pollination; consequently, the whitefly population increased on zucchini plants after cover removal. However, due to the larger size of the zucchini plants at anthesis, the subsequent negative impact of whiteflies on zucchini yields was lessened compared to exposure immediately after transplanting or seedling emergence.

Whitefly populations affected the dry biomass accumulation during the growing season at all locations. High whitefly populations in the early season in Georgia 2020 likely led to a lower dry biomass accumulation compared to that at other locations, regardless of the mulching treatment. However, the reflective silver plastic mulching and insect row cover treatments maximized the total dry biomass accumulation in Georgia 2020 and Georgia 2021. In Georgia 2020, total dry biomass accumulation was 246.3 kg ha^−1^ higher for the reflective silver plastic mulching than for the white plastic mulching, and 426.9 kg ha^−1^ higher for the cover treatment than the no-cover treatment. In Georgia 2021, total dry biomass accumulation was 605.6 kg ha^−1^ higher for the reflective silver plastic mulching than for the white plastic mulching, and 130.8 kg ha^−1^ comparing cover to no-cover treatments. In Alabama 2021, using reflective silver plastic and row covers resulted in a greater crop biomass accumulation rate constant and reduced the number of days to half biomass accumulation. The effect of plastic mulch and row cover treatments on zucchini plant growth may be related to several factors in addition to whitefly populations. Previous studies testing different colors of mulches to protect against whiteflies in tomatoes indicated that the differences in plant height may be caused by the effects of blue light on plants under high and low light intensities and by the effects of the increased far-red to red (FR/R) light ratio of the blue mulch on the seedlings, where, under relatively low light intensity, the FR/R ratio increased and resulted in increased plant height [13]. Warmer temperatures and increased plant transpiration for plants grown under row covers may have also contributed to superior dry biomass accumulation [31,34]. 

The differences among the treatments for dry biomass accumulation corresponded to the differences in total yields. Plants grown under reflective silver plastic mulching had a 17% average increase in total yield compared to those under white plastic mulching, while the insect row cover treatment increased zucchini total yields by 14% compared to those in the no-cover treatment. These results suggest that utilizing multiple cultural practices may reduce whitefly populations and increase total yields but could be an additional cost to the grower, and the economic viability should be considered in future studies. Combining reflective silver plastic mulching and insect row covering could be sustainable pest management practices for controlling whiteflies in zucchini production in the southeastern U.S. while reducing broad-spectrum pesticide use. These practices can be considered IPM practices, since row covers provide a barrier to whiteflies in early season, while silver plastic mulching may assist in repelling whiteflies until the expanded plant foliage covers the mulch.

## 5. Conclusions

Sweet potato whitefly is currently the main challenge facing vegetable production in the southeastern U.S. during the fall growing season. This is mostly due to the insect directly damaging vegetable crops and its role as a vector of many plant viruses. This study indicated that insect row cover and silver reflective mulch treatments decreased whitefly populations, increased dry biomass accumulation, and enhanced total yields compared to those in the no-cover and standard white plastic mulching treatments at all locations. Zucchini yield increased by 17% using silver reflective mulching and by 14% with insect row cover. Prior research emphasizes the importance of combining different management practices to reduce whitefly populations, and the present study corroborates the findings in the literature. This study introduces new integrated pest management practices for whitefly management in the southeastern U.S. Future studies can be conducted to evaluate the influence of cultural practices on the severity of whitefly vector viruses and whitefly’s natural enemies, which were not observed in this study.

## Figures and Tables

**Figure 1 insects-14-00863-f001:**
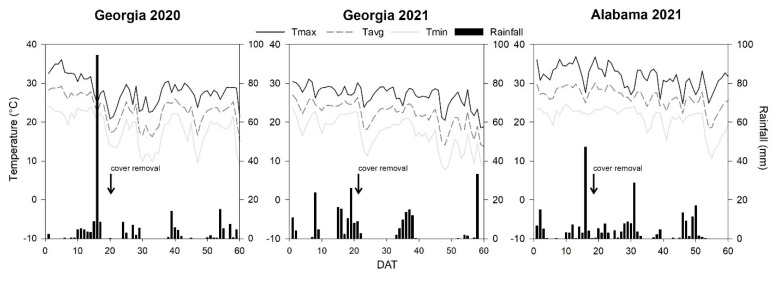
Weather conditions comprising daily rainfall events and maximum (Tmax), average (Tavg), and minimum (Tmin) air temperature during the zucchini growing season in Georgia 2020, Georgia 2021, and Alabama 2021.

**Figure 2 insects-14-00863-f002:**
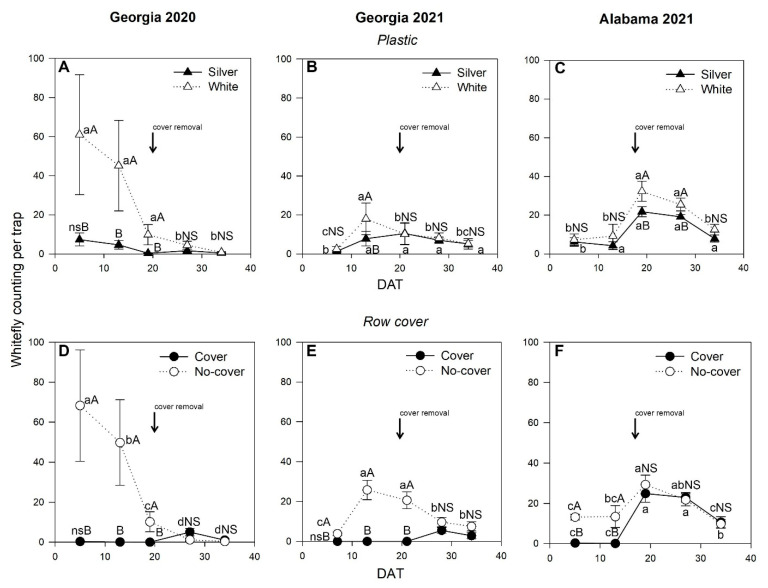
Effect of plastic mulching treatment within sampling time on whitefly population during the zucchini growing season for Georgia 2020 (**A**), Georgia 2021 (**B**), and Alabama 2021 (**C**), and row cover treatments within sampling time for Georgia 2020 (**D**), Georgia 2021 (**E**), and Alabama 2021 (**F**). Note: Different uppercase letters indicate significant differences (*p* ≤ 0.05) among plastic mulching and row cover treatments within sampling time according to Tukey’s mean test. According to Tukey’s mean test, different lowercase letters indicate a significant difference (*p* ≤ 0.05) among sampling time within plastic mulching or row cover treatments. ns and NS indicate no significant difference.

**Figure 3 insects-14-00863-f003:**
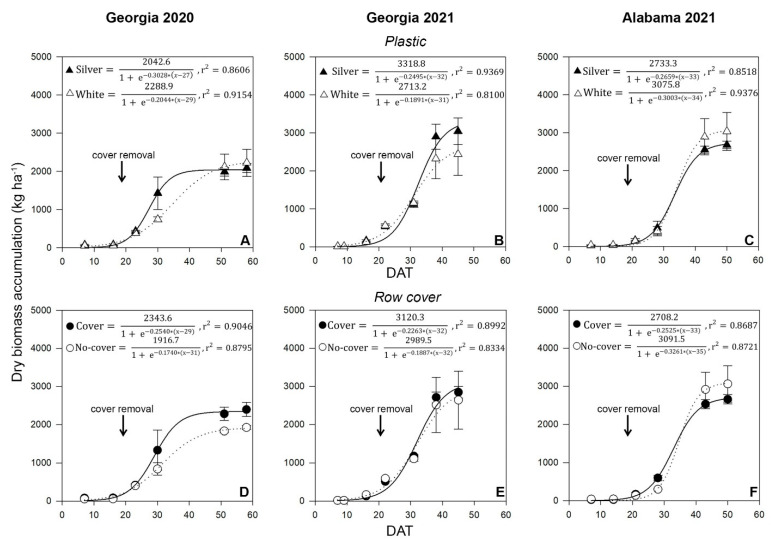
Effect of plastic mulch treatments for Georgia 2020 (**A**), Georgia 2021 (**B**), and Alabama 2021 (**C**) and row cover treatments for Georgia 2020 (**D**), Georgia 2021 (**E**), and Alabama 2021 (**F**) on biomass accumulation during the zucchini growing season.

**Table 1 insects-14-00863-t001:** The main effect of location, plastic color mulch, and row cover treatments and their interaction on total zucchini yield (kg ha^−1^).

Effects	Total Yield (kg ha^−1^)
Location	
Georgia 2020	11,451 b ^†^
Georgia 2021	15,177 a
Alabama 2021	15,248 a
*p*-value	**
Plastic mulching treatment	
Silver	15,245 a
White	12,672 b
*p*-value	*
Cover treatment	
Row cover	15,030 a
No-cover	12,887 b
*p*-value	*
Location × plastic mulching	
*p*-value	ns
Location × cover	
*p*-value	ns
Plastic mulching × cover	
*p*-value	ns
Location × plastic mulching × cover	
*p*-value	ns

ns, *, and **: nonsignificant or significant at *p* ≤ 0.05 or 0.01, respectively. ^†^ According to Tukey’s mean test, values followed by different letters indicate a significant difference (*p* ≤ 0.05) among treatments.

## Data Availability

The data presented in this study are available on request from the corresponding author.

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
