# Peer review of "Use of Insect Exclusion Row Cover and Reflective Silver Plastic Mulching to Manage Whitefly in Zucchini Production"

_insects, 2023, doi:10.3390/insects14110863_

Round 1
Reviewer 1 Report
Comments and Suggestions for Authors
T1. This kind of a study would have been much benefited by a cost comparison among treatments i.e. between the chemical applications and the cost of the covers in early season. The authors may add data if existed or add a comment on this. It is likely too costly to suggest combining both methods and environmentally risky (l. 317). A comment on the fate of the plastic can be also added. Effects on the whitely numbers may not be much improved since they are already strong, but effects on the plant development could be significant if both methods had been combined.
2. The density of the traps has to be given (l. 129).
3. Data on the whitefly eggs, nymphs and adults on the plants would be useful to follow the population development from the beginning per treatment.
4. Likely natural enemies of Bemisia tabaci, other pests or diseases may have appeared during the season or have different numbers or infection rates per treatment and biased the results. Data or at least a comment to clarify the situation to help the reader to assess the efficacy of the methods should be added.
5. The authors must compare the two methods and suggest the best one. The cover gives much more consistent results in all the three cases in regard to the whitefly population control early in the season.
6. The conclusions section is very long, be more concise, 5-6 lines would be fine, without repeating results. Effects on disease development, natural enemies, cost or other aspects can be suggested for near future research before use of the methods.
Author Response
1 This kind of a study would have been much benefited by a cost comparison among treatments i.e. between the chemical applications and the cost of the covers in early season. The authors may add data if existed or add a comment on this. It is likely too costly to suggest combining both methods and environmentally risky (l. 317). A comment on the fate of the plastic can be also added. Effects on the whitely numbers may not be much improved since they are already strong, but effects on the plant development could be significant if both methods had been combined.
Answer: The number of sprays is one of the benefits of the usage of those alternative managements. However, we did not specifically collect data on economic viability of treatments. Authors understand the point of the reviewer and we will consider future studies to evaluate this point. We added a statement at line 317 indicating needs for further studies.
2 The density of the traps has to be given (l. 129).
Answer: Authors added the information as requested, and it is highlighted in the manuscript.
3 Data on the whitefly eggs, nymphs and adults on the plants would be useful to follow the population development from the beginning per treatment.
Answer: Authors agree with the reviewer; however, we do not have the data on early stages of the insect.
4 Likely natural enemies of Bemisia tabaci, other pests or diseases may have appeared during the season or have different numbers or infection rates per treatment and biased the results. Data or at least a comment to clarify the situation to help the reader to assess the efficacy of the methods should be added.
Answer: Authors did not collect the suggested data, despite all monitoring and control of pests/disease was made as stated in the material and methods section. We will consider including those evaluations in future studies.
5 The authors must compare the two methods and suggest the best one. The cover gives much more consistent results in all the three cases in regard to the whitefly population control early in the season.
Answer: This comparison was made through the interaction presented in Table 1, and we identified that there was no significant interaction between treatment, only the main effect of each individual treatment. This result means both strategies reduced the impact of the whitefly on zucchini yield independntly, justifying the use of both treatments at the same time to improve crop yield.
6 The conclusions section is very long, be more concise, 5-6 lines would be fine, without repeating results. Effects on disease development, natural enemies, cost or other aspects can be suggested for near future research before use of the methods.
Answer: Authors shortened the conclusion section and added the suggestion.
Reviewer 2 Report
Comments and Suggestions for Authors
The manuscript is a standard research paper. The need for control measures against whitefly other than the usual chemical control method with its limitations (environmental damage, resistance) was identified and possible alternatives such as using a cover over the plants and a alight reflecting mulch were investigated. Statistically sound experiments were conducted and the results, which indicated positive control, discussed and brought into context with current knowledge, using references. The illustrations look good and are part of a well written manuscript. I have seldomly received a manuscript where so little changes are required and I have no hesitation in recommending acceptance .
You produced an excellent paper with interesting, useful results. I was particularly pleased that you considered factors other than whitefly that could have contributed to the differeces in biomass accumulation and yield.
A few suggestions: Line16, write challenges. Line 67: write ...control. In addition,....Line 333,write ...literature. Furthermore, ....
Use italics for scientific names in the abstract, unless editorial policy dictates otherwise.
Consider changing the title to The use of insect excluding row cover......The original did not make sence to me.
Author Response
The manuscript is a standard research paper. The need for control measures against whitefly other than the usual chemical control method with its limitations (environmental damage, resistance) was identified and possible alternatives such as using a cover over the plants and a alight reflecting mulch were investigated. Statistically sound experiments were conducted and the results, which indicated positive control, discussed and brought into context with current knowledge, using references. The illustrations look good and are part of a well written manuscript. I have seldomly received a manuscript where so little changes are required and I have no hesitation in recommending acceptance .
Answer: Thank you for the feedback.
You produced an excellent paper with interesting, useful results. I was particularly pleased that you considered factors other than whitefly that could have contributed to the differeces in biomass accumulation and yield.
Answer: Authors appreciate the observations.
A few suggestions:
Line16, write challenges. Line 67: write ...control. In addition,....Line 333,write ...literature. Furthermore, ....
Answer: Authors corrected all requested terms on the manuscript.
Use italics for scientific names in the abstract, unless editorial policy dictates otherwise.
Answer: Authors corrected the scientific name in the abstract.
Consider changing the title to The use of insect excluding row cover......The original did not make sence to me.
Answer: Authors changed the title adding an additional term to improve the understanding.
Reviewer 3 Report
Comments and Suggestions for Authors
Dear authors,
I would like to congratulate you on the excellent work presented in this study. Your research on efficient methods for controlling a troublesome insect species that affects economically important crops and demonstrates resistance to various insecticides is of great significance. Your findings regarding the use of silver reflective plastic mulching as an alternative for whitefly management in zucchini fields in the southeastern United States are noteworthy.
A few final remarks:
The clarity of your objectives and the experimental methodology are strengths of this study. The use of multiple locations for evaluation enhances the robustness of your conclusions.
The results are very promising, indicating that the row coverings for insects and silver reflective plastic mulching reduced whitefly populations, increased dry biomass accumulation, and improved overall yields. These findings have the potential to positively impact the agricultural industry and food security.
It is important to consider the practical implications of your work. How can this be implemented by farmers and influence existing agricultural practices? Exploring the practical and economic aspects of this method would be a valuable next step.
I encourage you to continue exploring this line of research and delve into related areas. Perhaps, further investigating the resilience of whiteflies to different control methods and exploring potential adaptations of the proposed techniques in varied climates may be a future path.
In summary, your work is a significant contribution to the field of agriculture and pest control. I hope you continue to build upon these promising results and share your knowledge with the agricultural community. Thank you for your effort and dedication to research that benefits our society.
Sincerely,
Comments on the Quality of English Language
He does not have enough expertise to evaluate, but I inform him that his understanding is satisfactory.
Author Response
I would like to congratulate you on the excellent work presented in this study. Your research on efficient methods for controlling a troublesome insect species that affects economically important crops and demonstrates resistance to various insecticides is of great significance. Your findings regarding the use of silver reflective plastic mulching as an alternative for whitefly management in zucchini fields in the southeastern United States are noteworthy.
A few final remarks:
The clarity of your objectives and the experimental methodology are strengths of this study. The use of multiple locations for evaluation enhances the robustness of your conclusions.
The results are very promising, indicating that the row coverings for insects and silver reflective plastic mulching reduced whitefly populations, increased dry biomass accumulation, and improved overall yields. These findings have the potential to positively impact the agricultural industry and food security.
It is important to consider the practical implications of your work. How can this be implemented by farmers and influence existing agricultural practices? Exploring the practical and economic aspects of this method would be a valuable next step.
I encourage you to continue exploring this line of research and delve into related areas. Perhaps, further investigating the resilience of whiteflies to different control methods and exploring potential adaptations of the proposed techniques in varied climates may be a future path.
In summary, your work is a significant contribution to the field of agriculture and pest control. I hope you continue to build upon these promising results and share your knowledge with the agricultural community. Thank you for your effort and dedication to research that benefits our society.
Answer: Thank you for your considerations and suggestions. We will consider all those comments in our future work.